# A Scalable Cross-Chain Access Control and Identity Authentication Scheme

**DOI:** 10.3390/s23042000

**Published:** 2023-02-10

**Authors:** Yuhang Ding, Yanran Zhang, Bo Qin, Qin Wang, Zihan Yang, Wenchang Shi

**Affiliations:** 1Renmin University of China, Beijing 100872, China; 2CSIRO Data61, Sydney, NSW 2131, Australia

**Keywords:** blockchain, cross-chain, access control, identity authentication

## Abstract

Cross-chain is an emerging blockchain technology which builds the bridge across homogeneous and heterogeneous blockchains. However, due to the differentiation of different blockchains and the lack of access control and identity authentication of cross-chain operation subjects, existing cross-chain technologies are struggling to accomplish the identity transformation of cross-chain subjects between different chains, and also pose great challenges in terms of the traceability and supervision of dangerous transactions. To address the above issues, this paper proposes a scalable cross-chain access control and identity authentication scheme, which can authenticate the legitimacy of blockchains in the cross-chain system and ensure that all cross-chain operations are carried out by verified users. Furthermore, it will record all cross-chain operations with the help of Superchain in order to regulate and trace illegal transactions. Our scheme is scalable and, at the same time, has low invasiveness to blockchains in the cross-chain system. We implement the scheme and accordingly conduct the evaluations, which prove its security, efficiency, and scalability.

## 1. Introduction

Blockchain [1] is a decentralized distributed ledger which establishes trust between different subjects in the blockchain network with the help of technologies such as cryptography, distributed networks, and consensus mechanisms. This unique feature enables each node in the blockchain to process transactions more transparently and reliably in a trusted network. At the same time, the tamper-proof feature guaranteed by cryptography technology also enables the recording and tracing of suspicious transactions. Therefore, blockchain technology has attracted widespread attention from all walks of life, and blockchains for various application scenarios are constantly emerging [2]. With the increasingly complex scenarios of blockchain applications, solving the problem of data islands between different chains and realizing data circulation among different blockchains have become urgent demands [3], and cross-chain technologies have emerged as the times require.

Cross-chain technology aims to connect homogeneous or heterogeneous blockchains so as to achieve interaction between different chains [4]. However, the existing cross-chain technologies have not adequately solved the problem of identity authentication and access control between different chains, especially public blockchains [5]. At present, the vast majority of blockchain nodes are running in servers in the territory of sovereign countries, and some permission blockchains even have nodes running in the jurisdiction of a sovereign country [6], which makes them subject to the constraints and supervision of the country’s laws and policies. At the same time, different countries or regions have different regulatory policies for cross-border data circulation [7]. Take cross-border payment as an example, according to Chapter 31 of the United States Code on the import and export of monetary instruments [8], when a monetary instrument with a value of more than USD 10,000 passes through the United States and beyond the United States (including the starting point and destination), any person or their agent or trustee with the knowledge shall submit a report at the time and place that the Secretary of the Treasury prescribes, which shall contain the following information: (i) the legal capacity in which the person filing the report is acting; (ii) the origin, destination, and route of the monetary instruments; (iii) when the monetary instruments are not owned or used by the person transporting the instruments, the identity of one of the sending and receiving parties or both are required; (iv) the amount and kind of monetary instruments transported; and (v) additional information. Similarly, China [9] and the European Union [10] have also formulated laws and regulations on this issue with different contents but the same purpose. Therefore, in the foreseeable future, in order to adapt to the regulations of different countries on on-chain transactions and to trace illegal cross-chain transactions, it will become a necessary measure to authenticate the authority and identity of the source chain and users who initiate cross chain access.

Existing cross-chain systems are mainly based on notary scheme [3], hash-locking [11], sidechain [12,13] or relay chain [14,15]. Among them, the scheme based on relay chains has more application scenarios and better scalability. However, as far as we know, the existing cross-chain schemes have not realized the identity authentication and access control of participating nodes, as well as the supervision and traceability of cross-chain transactions in practice. This poses a huge challenge to financial regulation and compliance. Therefore, the focus of this paper is to design a set of available, highly scalable and low intrusive cross-chain system with identity authentication and access control, so as to provide regulatory protection for future cross-border cross-chain contract interaction and transaction circulation. The work is mainly faced with the following challenges:aAchieving transaction circulation between heterogeneous blockchains with different chain data structures, transaction order formats, and consensus algorithms while maintaining low chain intrusion;bAchieving identity authentication and access control for different nodes and users from different chains. Moreover, achieving the unification of cross-chain system identity authentication and access control under different chain management rules;cAchieving the connection of different chains without modifying the underlying code of the application chain, that is, with limited development freedom, while maintaining a low chain intrusion.

Based on the above problems, we propose an extensible cross-chain access control and identity authentication scheme, which can authenticate the legitimacy of the chains in the cross-chain system and ensure that all cross-chain operations are carried out by verified users. Furthermore, it will record all cross-chain operations with the help of Superchain in order to regulate and trace illegal transactions. This proposed scheme is easy to deploy in a real scenario. The main contributions of this paper are summarized as follows.

-To solve challenge **a**, we improve the relay chain-based cross-chain framework and propose an improved scheme with high scalability and low intrusion into participating chains.-To solve the challenge **b**, we design a cross-chain access control and identity authentication scheme, which can realize cross-chain identity conversion between different chains and record and trace illegal transactions.-To solve challenge **c**, we implement experiments with our proposed cross-chain framework and identity authentication and access control scheme, which proves its security, efficiency, and scalability.

The rest of this paper is organized as follows. Section 2 introduces the related work on cross-chain technologies. Section 4 introduces the components and the basic procedures of our cross-chain system, as well as the notations we use. Section 5 presents an extensible new cross-chain access control and identity authentication scheme. In Section 6, we evaluate our cross-chain access control and identity authentication scheme through experiments. Section 7 concludes this work.

## 2. Related Work

In this section, we provide an overview of existing studies on cross-chain technologies. Existing popular cross-chain technologies include four mainstream solutions: notary scheme [3], hash-locking [11], sidechain [12,13], and relay chain [14,15]. We compared the differences and main pros and cons between the references in Table 1.

### 2.1. Notary Scheme

The notary scheme [3] is a cross-chain scheme that simply introduces a trusted third party, which acts as a notary through a single independent node or distributed nodes to verify the legitimacy and consistency of cross-chain transactions, which is easy to implement in a real scenario. One of the most famous notary scheme-based cross-chain solutions is the Interledger protocol, proposed by S. Thomas in 2012 [16], which is applied to Ripple [17]. However, the notary scheme has the risk of being attacked by evil notaries, and is unable to interact with smart contracts across chains. Some cross-chain solutions try to resort to cryptography algorithms such as multi-signature to resolve the centralized risk [17,18], however, it is just a delaying tactic, which still has the possibility of being attacked by a third evil party.

### 2.2. Hash-Locking

Hash-locking is also called Hashed-Timelock Agreements, which first appeared in the lightning network and was originally designed to solve the scalability problem of Bitcoin by Poon in 2015 [19,20]. Hashed-Timelock requires both parties to provide the corresponding voucher within the agreed time, and the submitted voucher is the correct preimage of the hash function [24]. When conducting cross-chain transactions, both parties of the transaction can lock assets, setting the corresponding time and unlocking conditions through communication without the intervention of a third party to realize the atomic swap. However, this scheme can only be implemented when both parties are online at the same time, which limits the application scenarios.

### 2.3. Sidechain

The sidechain technology was first defined by BlockStream [13] in 2014. Then, Gaži [12] further formalized it by proposing a rigorous cryptographic definition, which mainly uses bidirectional pegging, one way to realize the circulation of assets between the main chain and the side chain, to realize asset exchange and data circulation between different chains. The sidechain technology is able to relieve the pressure on the main chain and can store and process a portion of the transactions alone. However, sidechain technology has increased system complexity and introduced new security problems, such as fraudulent transfer and mining centralization [25].

### 2.4. Relay Chain

The relay chain [14,15] connects different chains by constructing the cross-chain message-passing protocols in the cross-chain system through the third chain, and realizes the data circulation and state verification between different chains. All cross-chain operations from each chain in the cross-chain system will be recorded on the relay chain and verified by it. The relay chain solution is suitable for cross-chain interaction between heterogeneous chains with different communication standards and consensus algorithms and is highly scalable [21]. There are a large number of mature projects that choose to use relay chain as cross-chain solutions. In 2019, Cosmos [22] proposes the Inter-Blockchain Communication protocol for cross-chain interaction based on relay chains. Same as Cosmos, Polkadot [23] has also applied this solution.

## 3. Background

In this section, we provide the background information surrounding our proposed scheme. We introduce the concept of both blockchain and cross-chain.

**Blockchain.** The blockchain technology was first proposed by Nakamoto [1] and is maintained by distributed nodes. Its manifestation is a chain structure composed of different blocks logically connected by hash values. Under common conditions, each block includes the block header and the block body. The block header mainly includes the hash of the previous block, nonce and other information of the block, while the block body stores transactions and transaction verification signatures. Nodes in the blockchain system participate in the competition of block generation rights according to the consensus algorithm and receive token rewards. However, due to the difference in transaction order data structure, block data structure, chain data structure, and consensus algorithms of different chains, transactions between heterogeneous blockchain systems cannot be directly transferred. This makes different blockchain systems an isolated island of data, greatly limiting the application scenarios of blockchain.

**Cross-Chain Technology.** Cross-chain technology has been of great interest since the emergence of blockchain. The first cross-chain technology to be realized in a real sense was the Interledger[16] protocol proposed by Ripple Labs in 2012. The initial cross-chain technology only focused on asset exchange between different chains. However, with the development of blockchain and the great success of smart contracts, the interaction between users and smart contracts became another important activity in the blockchain after the exchange of virtual assets. However, the interaction of cross-chain smart contracts is more complex than ordinary asset exchange, because transactions generated through smart contracts often contain more complex information than transfer transactions, so it is difficult to achieve through the notary scheme and hash locking scheme. Therefore, in order to overcome the differences in network topology, data structure, consensus algorithm, block and transaction generation and verification logic between different blockchains, relay chain, and sidechain schemes became mainstream methods to understand the cross-chain interaction of smart contracts. The sidechain scheme mainly focuses on the interworking of isomorphic blockchains and has low scalability. Relaychain schemes are often highly invasive to the chains involved in cross-chain systems, and the implementation is relatively complex.

## 4. Cross-Chain System

In this section, we will first define the notations we used in this paper in Abbreviations and present the system model, as well as introduce the components and the basic framework of our cross-chain system.

### 4.1. Components

We design a new cross-chain framework based on the relay chain scheme, which is composed of application chains, Superchain, and auto agents.

#### 4.1.1. Application Chains

Application chains are the participants of the cross-chain system, and users on the application chains play the role of senders and receivers of the cross-chain operations in the system. For the application chain which supports smart contracts, each has a cross-chain smart contract that interacts with users, which can realize the identification of cross-chain operations from users and transfer them to auto agents. For the application chains which do not support smart contracts, they can only realize the simple function of transferring accounts with the help of auto agents.

#### 4.1.2. Auto Agent

Auto agent is a virtual user abstracted from each chain, which plays the role of a super chain node and application chain node in the cross-chain network. Each application chain can have multiple auto agents and form an auto agent committee, which can repackage the transactions from the application chain into the data format of the Superchain’s transaction and can also repackage the transactions from the Superchain into the corresponding data format of the application chain’s transaction, thus realizing the transaction circulation in the cross-chain system.

#### 4.1.3. Superchain

Superchain plays the role of a relay chain in the cross-chain system. Its main function is to store certificates and record users’ cross-chain operations for supervision and when cross-chain operations are performed through the Superchain, the Superchain will verify both parties involved in the cross-chain operations. For the Superchain, its users are the application chains in the lower layer, and each application chain has several certificates of Superchain users. In the system, the actual users of the Superchain are the auto agent nodes of each application chain.

### 4.2. Framework

When an application chain newly participates in the cross-chain system, some nodes of the application chain will run the client of the Superchain locally, becoming a node of the Superchain, and then complete the process of chain registration. At this time, this node becomes an auto agent of the application chain, which is responsible for the cross-chain data circulation of the chain. As shown in Figure 1, the detailed processes are as follows:

① *u* invokes the cross-chain smart contract of $AC_{i}$ to generate a cross-chain transaction $tx_{i}$ from $AC_{i}$ to $AC_{j}$ with parameters <FromChainID,ToChainID,Options>.

② Each node in $AC_{i}$ broadcasts the transaction $tx_{i}$.

③ $AGRobot_{i}$ repackages $tx_{i}$ into the format of SC, which turns into $tx_{s}$, then $AG_{i}$ broadcasts $tx_{s}$ in SC.

④ $AG_{j}$ receives $tx_{s}$ and confirms $ToChainID=ChainID_{j}$.

⑤ $AGRobot_{j}$ repackages $tx_{s}$ into the format of $AC_{j}$, which turns into $tx_{j}$, and then $AG_{j}$ broadcasts $tx_{j}$ in $AC_{j}$ and returns the results.

In the following, we will introduce the detailed process of chain registration and cross-chain data circulation in two parts:

#### 4.2.1. Chain Registration

Chain registration is a necessary preparation process for each application chain before joining the cross-chain system. Its main purpose is to generate auto agents and establish the connection between the Superchain and the application chain. Assume that the cross-chain system has n+1 presently chains, which means that $CHAINS=\{SC,AC_{0},AC_{1},\cdots ,AC_{n-1}\}$ and each application chain $AC_{i}$ has a committee composed of auto agents which are represented by $AG_{i}$, where i={0,1,⋯,n−1}.

When an application chain $AC_{n}$ wants to participate in the cross-chain system, some nodes in $AC_{n}$ need to run the client of Superchain SC locally, and they will become nodes of SC. We name these special dual nodes as auto agents of $AC_{n}$, which are represented by $AG_{n}^{j}$, where *j* is less than the number of dual nodes. $AG_{n}^{j}$ have addresses in both $AC_{n}$ and SC, respectively, which are represented by $AGACaddr_{i}^{j}$ and $AGSCaddr_{i}^{j}$. To overcome the heterogeneity between the application chain and Superchain, it also needs to run a transaction processing program $AGRobot_{i}$, which is independent of the blockchain clients on the server of $AG_{i}$ (notice that this design does not change the code of clients). $AGRobot_{i}$ can identify the transaction format of the application chain and Superchain, and can repackage the transactions according to the destination chain’s transaction format of the transactions.

As shown in Figure 2, the blue nodes are the normal nodes of $AC_{n}$, the pink nodes are the normal nodes of $AC_{i}$, and the green nodes are the normal nodes of SC. When $AC_{n}$ participates in the cross-chain system, some normal nodes will become auto agents and they are filled with blue-green gradients, as is $AC_{i}$. At this time, |CHAINS|=n+2 and $CHAINS=\{SC,AC_{0},AC_{1},\cdots ,AC_{n-1},AC_{n}\}$.

#### 4.2.2. Cross-Chain Data Circulation

Cross-chain transactions are generated by the users of each application chain by invoking the cross-chain contract of the chain. Assume that $CHAINS=\{SC,AC_{0},AC_{1},\cdots ,AC_{n-1},$$AC_{n}\}$ and $AC_{i},AC_{j}\in CHAINS$.

When user *u* of application chain $AC_{i}$ invokes a cross-chain contract to access application chain $AC_{j}$, it will generate a transaction with the contract address as the target address. The transaction saves the values of the parameters inputted by the user into the contract, which is recorded by $<FromChainID:ChainID_{i},ToChainID:ChainID_{j},Options:Options_{i}^{j}>$. According to the transaction broadcasting rules of the blockchain, almost all honest nodes in $AC_{i}$ will receive and broadcast this transaction (when there is a good network condition), execute the corresponding smart contract code according to the content of the transaction, and save the latest status of the smart contract. In particular, when $AG_{i}$ receives a transaction in which the target address is the address of the cross-chain contract, it will check whether the FromChainID contained in the transaction is the same as ToChainID. If FromChainID=ToChainID, the transaction is not a cross-chain transaction and $AG_{i}$ will conduct routine processing according to the normal chain transaction. If FromChainID≠ToChainID, it indicates that the target chain of this transaction is not $AC_{i}$, which means that it is a cross-chain transaction. At this time, with the help of $AGRobot_{i}$, the auto agent will repackage the transaction into the format of the Superchain’s transaction, and broadcast the packaged transaction in Superchain. Similar to the above process, when the network is in good condition, all honest nodes in the Superchain will receive and broadcast the transaction. When auto agent node $AG_{k}$ receives the transaction, where 0≤k≤n, it will check whether the ToChainID contained in the transaction is the same as $ChainID_{k}$. If $ToChainID\neq ChainID_{k}$, $AG_{k}$ will conduct routine processing according to the normal Superchain transaction. If $ToChainID=ChainID_{k}$, which means that the transaction is a cross-chain transaction that points to $AC_{k}$, the auto agent will repackage the transaction into the format of $AC_{k}$’s transaction, and broadcast the packaged transaction in $AC_{k}$. Then, the nodes of $AC_{k}$ execute the corresponding smart contract code according to the transaction and save the latest status of the smart contract. The cross-chain data circulation process is shown in detail in Algorithm 1.
**Algorithm 1** Cross-chain data circulation1:**Input:**2:   CHAINS: the set of chains in the cross-chain system3:   AG: the set of auto agents4:   AGRobot: the set of AGRobots5:   CrossCon: the set of cross-chain contracts6:**Output:**7:   True or False8:————————————————————————————————————————9:**1. Propose ($u_{i}$ from $AC_{i}$ to $AC_{j}$):**10:   $FromChainID\leftarrow ChainID_{i}$11:   $ToChainID\leftarrow ChainID_{j}$12:   $tx_{i}\leftarrow CrossCon_{i}\left(FromChainID,ToChainID,Options\right)$13:   Broadcast $tx_{i}$ in $AC_{i}$14:   **invoke** (Cross-chain broadcast)15:- - - - - - - - - - - - - - - - - - - - - - - - - - - - - - - - - - - - - - - - - - - - - - - - - - - - - - - - - - - - - - - - - -16:**2. Cross-chain broadcast:**17:   $AG_{i}$ receives $tx_{i}$18:   **if** FromChainID≠ToChainID **then**19:      $tx_{s}\leftarrow AGRobot\left(tx_{i}\right)$20:      Broadcast $tx_{s}$ in SC21:      **invoke** (Response)22:   **else**23:      Broadcast $tx_{i}$ in $AC_{i}$24:      return false25:- - - - - - - - - - - - - - - - - - - - - - - - - - - - - - - - - - - - - - - - - - - - - – - - - - - - - - - - - - - - - - - - -26:**3. Response ($AG_{k}$ receives $tx_{s}$):**27:   **if** $ToChainID=ChainID_{k}$ **then**28:      $tx_{k}\leftarrow AGRobot\left(tx_{s}\right)$29:      Broadcast $tx_{k}$ in $AC_{k}$30:      $CrossCon_{k}\left(FromChainID,ToChainID,Options\right)$31:      Return True32:   **else**33:      Broadcast $tx_{s}$ in SC34:      Return False

## 5. Cross-Chain Access Control and Identity Authentication Scheme

In this section, we will present an extensible chain-level cross-chain access control and identity authentication scheme based on the above cross-chain system, which can balance the efficiency and security of the system. As follows, we will first introduce this scheme by improving the processes of chain registration and cross-chain data circulation, before analyzing the scheme in terms of security and advanced properties. The cross-chain data circulation process is shown in Algorithm 2 in detail.
**Algorithm 2** Cross-chain data circulation with access control and identity authentication (I: improvement point)1:**Input:**2:   CHAINS: the set of chains in the cross-chain system3:   AG: the set of auto agents4:   AGRobot: the set of AGRobots5:   CrossCon: the set of cross-chain contracts6:**Output:**7:   True or False8:————————————————————————————————————————9:**1. Propose ($u_{i}$ from $AC_{i}$ to $AC_{j}$):**10:   $FromChainID\leftarrow ChainID_{i}$11:   $ToChainID\leftarrow ChainID_{j}$12:   Authenticate $u_{i}$ in $AC_{i}$                         ▹I13:   $tx_{i}\leftarrow CrossCon_{i}\left(FromChainID,ToChainID,Options\right)$14:   Broadcast $tx_{i}$ in $AC_{i}$15:   **invoke** (Cross-chain broadcast)16:- - - - - - - - - - - - - - - - - - - - - - - - - - - - - - - - - - - - - - - - - - - - - – - - - - - - - - - - - - - - - - - - -17:**2. Cross-chain broadcast:**18:   $AG_{i}$ receives $tx_{i}$19:   **if** FromChainID≠ToChainID **then**20:      $Sig_{AG_{i}}\leftarrow SK_{AG_{i}}\left(ToChainID\right)$                   ▹I21:      $ACIAC_{s}\leftarrow \left(FromChainID,ToChainID,Sig_{AG_{i}}\right)$         ▹I22:      **if** $PK_{AG_{i}}\left(Sig_{AG_{i}}\right)\equiv ToChainID$
**then**                ▹I23:         $tx_{s}\leftarrow AGRobot\left(tx_{i}\right)$24:         Broadcast $tx_{s}$ in SC25:         **invoke** (Response)26:      **else**                                  ▹I27:         Return False                               ▹I28:   **else**29:      Broadcast $tx_{i}$ in $AC_{i}$30:      Return False31:- - - - - - - - - - - - - - - - - - - - - - - - - - - - - - - - - - - - - - - - - - - - - – - - - - - - - - - - - - - - - - - - -32:**3. Response ($AG_{k}$ receives $tx_{s}$):**33:   **if** $ToChainID=ChainID_{k}$ **then**34:      $Sig_{AG_{j}}\leftarrow SK_{AG_{j}}\left(FromChainID\right)$                   ▹I35:      $ACIAC_{s}\leftarrow \left(FromChainID,ToChainID,Sig_{AG_{j}}\right)$          ▹I36:      **if** $PK_{AG_{j}}\left(Sig_{AG_{j}}\right)\equiv FromChainID$
**then**                ▹I37:         $tx_{k}\leftarrow AGRobot\left(tx_{s}\right)$38:         Broadcast $tx_{k}$ in $AC_{k}$39:      **else**40:         Return False                              ▹I41:      $CrossCon_{k}\left(FromChainID,ToChainID,Options\right)$42:      Return True43:   **else**44:      Broadcast $tx_{s}$ in SC45:      Return False

### 5.1. Chain Registration with Access Control and Identity Authentication

The main process of chain registration of this scheme is similar to that in Section 4. Each application chain $AC_{i}$ has a group of auto agents $AG_{i}$, which includes both the client of $AC_{i}$ and the client of SC. Therefore, $AG_{i}$ can create user $u_{i}$ of $AC_{i}$ and $u_{s}$ of SC and both $<PK_{u_{i}},SK_{u_{i}}>$ and $<PK_{u_{s}},SK_{u_{s}}>$ are visible to $AGRobot_{i}$. In order to achieve the access control and identity authentication of $AG_{i}$, the system needs to carry out the following extra processes to achieve mutual authentication between $AC_{i}$ and SC.

(1)SC deploys an access control and identity authentication contract ($ACIAC_{s}$) to store the $PK_{u_{i}}$ of each $AG_{i}$ and $AG_{i}$ nodes can only have one user, which behavior rules are specified by codes. Similarly, each $AC_{i}$ needs to deploy an $ACIAC_{i}$ to store the $PK_{u_{s}}$ of each $AG_{i}$.(2)Each $AG_{i}$ invokes the $ACIAC_{s}$ with parameters $<AGSCaddr_{s}^{j},PK_{u_{i}}>$ to store the address of $AG_{i}$ in SC and the public key of $AG_{i}$ in $AC_{i}$ and invokes the $ACIAC_{i}$ with parameters $<AGACaddr_{i}^{j},PK_{u_{s}}>$ to store the address of $AG_{i}$ in $AC_{i}$ and the public key of $AG_{i}$ in SC.

As shown in Figure 3, the blue nodes belong to $AC_{i}$ and the green nodes belong to SC and $AG_{i}$ is formed by one $AC_{i}$’s node and one SC’s node. After the processes of chain registration with access control and identity authentication, $ACIAC_{i}$ has restored $PK_{u_{s}}$ and $AGACaddr_{i}$ and $ACIAC_{s}$ has restored $PK_{u}$, AGSCaddr and the corresponding ChainID, which realizes the registration of auto agents in Superchain and application chain.

### 5.2. Cross-Chain Data Circulation with Access Control and Identity Authentication

In order to achieve the access control and identity authentication of cross-chain transactions, we made some improvements to the process of cross-chain data circulation in Section 4, which requires authentication when receiving the cross-chain transactions sent by users or transferred by auto agents, so as to achieve access control. As shown in Figure 4, the detailed processes are as follows:

① *u* invokes the cross-chain smart contract of $AC_{i}$ to generate a cross-chain transaction $tx_{i}$ from $AC_{i}$ to $AC_{j}$ with parameters $<FromChainID,ToChainID,Options,Sig_{u}>$.

② $AC_{i}$ verifies $Sig_{u}$ according to its own user identity management scheme (this is not the cardinal part we care about in the cross-chain system, because each application chain has its own identity authentication and access control scheme, and we only care about how to authenticate in the cross-chain process [12]). Then, broadcast the transaction $tx_{i}$.

③ $AG_{i}$ receives $tx_{i}$, then invokes the cross-chain smart contract of SC to require the certificate of ToChianID with parameters $<FromChainID,ToChianID,Sig_{AG_{i}}>$, where $Sig_{AG_{i}}$ is the digital signature of this option calculated by $SK_{AG_{i}}$.

④ SC confirms that the certificate of $AG_{i}$ exists on the cross-chain smart contract, and uses $PK_{AG_{i}}$ to verify the signature $Sig_{AG_{i}}$, confirming that it is indeed generated using its own identity. After verifying the above information, SC replies to $AG_{i}$ with a query request about the certificate $<ChainID_{j},PK_{AG_{j}}>$ of $AG_{j}$.

⑤ $AGRobot_{i}$ repackages $tx_{i}$ to the format of SC, which turns into $tx_{s}$, then $AG_{i}$ broadcasts $tx_{s}$.

⑥ $AG_{j}$ receives $tx_{s}$.

⑦ $AG_{j}$ invokes the cross-chain smart contract of SC to require the certificate of ToChianID with parameters $<FromChainID,ToChianID,Sig_{AG_{j}}>$, where $Sig_{AG_{j}}$ is the digital signature of this option calculated by $SK_{AG_{j}}$.

⑧ SC confirms that the certificate of $AG_{j}$ exists on the cross-chain smart contract, and uses $PK_{AG_{j}}$ to verify the signature $Sig_{AG_{j}}$, confirming that it is indeed generated using its own identity. After verifying the above information, SC replies to $AG_{j}$ with a query request about the certificate $<ChainID_{i},PK_{AG_{i}}>$ of $AG_{i}$.

⑨ $AGRobot_{j}$ repackages $tx_{s}$ into the format of $AC_{j}$, which turns into $tx_{j}$, then $AG_{j}$ broadcasts $tx_{j}$ and returns the results.

### 5.3. Security Analysis

In this part, we will briefly argue the Sybil-resistance of the system referring to [26] and sketch the security analysis of our constructions.

**Adversary model:** We denote the adversary in our cross-chain system by A, which can statically and actively corrupt up to *t* of the *n* auto agents in AG, for t<<n.

**Definition** **1**(Sybil-resistance). *Let λ be the security parameter. A cross-chain system is Sybil-resistant with respect to a set of auto agents if, for any stateful PPT adversary A, $Pr[G^{sybil}\left(\lambda ,A,AG,tx\right)$⇒1]≤negl(λ).*

Informally, this definition points out that it is infeasible for an adversary to control the broadcasting of cross-chain transactions by controlling a limited number of auto agents. The definition is parameterized by the set of auto agents AG. Algorithm 3 specifies the game, where the adversary initializes *k* auto agents (k≤t<<n) and can participate in cross-chain data circulation. The adversary wins by forging the cross-chain transaction, tampering with transactions or withholding transactions to ensure that the target blockchain receives the cross-chain transactions that violate the real intention of the source blockchain.
**Algorithm 3** Sybil-resistant game $G^{sybil}$ (from $AG_{i}$ to SC)1:**Input:**2:   $AG_{i}$: the set of auto agents3:   $tx_{in}$: the transaction first receives by $AG_{i}$4:   A: the adversary5:   λ: the security parameter6:**Output:**:7:   $tx_{out}$: the transaction first transferred to SC by $AG_{i}$8:————————————————————————————————————————9:**Initial:**10:$PK_{AG_{i}},SK_{AG_{i}},PK_{u_{s}},SK_{u_{s}}\leftarrow Registration\left(1^{\lambda }\right)$11:A$init(AG_{A})$, where $|AG_{A}|=k$12:- - - - - - - - - - - - - - - - - - - - - - - - - - - - - - - - - - - - - - - - - - - - - – - - - - - - - - - - - - - - - - - - -13:**A.1. Forge the cross-chain transaction:**14:   $tx_{F},Sig_{AG_{i}}\left(tx_{F}\right)\leftarrow AG_{A}\left(SK_{AG_{i}}\right)$15:   **if** $Verify(Sig_{AG_{i}}\left(tx_{F}\right))$ **then**16:      Broadcast $tx_{F}$ in SC17:      **return** $tx_{out}=tx_{F}$18:   **else return** nil19:- - - - - - - - - - - - - - - - - - - - - - - - - - - - - - - - - - - - - - - - - - - - - – - - - - - - - - - - - - - - - - - - -20:**A.2. Tamper with the cross-chain transaction:**21:   $AG_{A}$ receives $tx_{in}$22:   $tx_{T},Sig_{AG_{i}}\left(tx_{T}\right)\leftarrow AG_{A}^{Tamper}\left(SK_{AG_{i}},tx_{in}\right)$23:   **if** $Verify(Sig_{AG_{i}}\left(tx_{T}\right))$ **then**24:      Broadcast $tx_{T}$ in SC25:      **return** $tx_{out}=tx_{T}$26:   **else return** nil27:- - - - - - - - - - - - - - - - - - - - - - - - - - - - - - - - - - - - - - - - - - - - - – - - - - - - - - - - - - - - - - - - -28:**A.3. Withhold the cross-chain transaction:**29:   $AG_{A}$ receives $tx_{in}$30:   withhold the transaction31:   $AG_{A}$ receives $tx_{in2}$, broadcast32:   **return** $tx_{out}=tx_{in2}$

**Theorem** **1.**
*The system is Sybil-resistant in the case that the adversary A does not have the advantage of network connectivity in either case of $G^{sybil}$.*


**Proof of Theorem** **1.**A controls *k* auto agents where k≤t<<n. We denote the set of corrupt auto agents as $AG_{A}$, where $|AG_{A}|=k$. Considering A.1 and A.2, for all of the auto agents have at least the same network connectivity as $AG_{A}$, the possibility that $AG_{A}$ takes the lead in broadcasting the $tx_{F}$ or $tx_{T}$ is k/n and the transaction initiator can execute another query after the transaction to protect against A.2. Similarly, A.3 can also be prevented by querying after the transaction. Therefore, the adversary cannot win the game in either case. □

### 5.4. Scalability

In this part, we will introduce the scalability of the cross-chain system with the access control and identity authentication scheme from the perspectives of application chains, nodes, and users.

#### 5.4.1. Application Chains

Our proposed cross-chain system with identity authentication and access control is an extensible system for application chains. When a new blockchain participates in the cross-chain system, it only needs to verify some nodes to run the Superchain client according to its internal access control and identity authentication rules, and run an AGRobot off the chain (the same kind of blockchain only needs to write one copy of AGRobot, that is, for the same kind of blockchain, AGRobot is reusable), which has low invasiveness to both Superchain and application chains.

#### 5.4.2. Nodes

There are two main types of nodes existing in our cross-chain system, namely regular nodes (including application chains and the Superchain) and auto agent nodes, both of which are scalable. Regular nodes only need to expand according to the rules of their own chain, and the joining and exiting of regular nodes do not logically conflict with the cross-chain system. For auto agent nodes, they need to complete the chain registration process when joining, i.e., it needs to run both the Superchain node and their own chain node, and complete the mutual authentication of the Superchain’s public key and their own chain’s public key. Finally, copying the AGRobot of other auto-agent nodes completes the extension of the auto-agent nodes.

#### 5.4.3. Users

Since this cross-chain scheme is less invasive to blockchains, it allows the creation of users on each node of the application chain according to the account management rules of these blockchains. If users want to initiate cross-chain transactions, they only need to invoke cross-chain contracts, thus enabling the scaling of users.

## 6. Implementation and Evaluation

### 6.1. Configuration

As a proof-of-concept, we implement a cross-chain system containing five blockchains, which are Ethereum, Hyperledger Fabric, Fisco BCOS, CITA, and Xuperchain, with the proposed access control and identity authentication scheme, and test the performance of this system. We conduct our experiment on six instances, each of which has two vCPUs with 8 GB of main memory installed and a 60 GB hard drive. The cluster of instances has a public network IP, using a springboard machine to connect the instances, and the instances communicate with each other through the internal network. The basic configuration of the experiment is shown in Table 2.

### 6.2. Experiment and Analysis

We designed the corresponding transaction parsing program AGRobot for each chain, which has good portability and can be directly added to a new auto-agent node to complete the repackaging of cross-chain transactions and stress-tested each of the five blockchains in the cross-chain system. Different auto agent nodes of the same application chain run both the client of this chain and the client of the Superchain, and communicate within the application chain through the communication system of the application chain itself, and similarly communicate between the Superchain nodes through the communication system of the Superchain. Therefore, for application chains, this way of accessing the cross-chain system does not have any impact on the architecture of the chain itself, which only needs to run a copy of the AGRobot program on the instance where the auto agent node is located.

In order to verify the scalability and performance of the system, we conducted experiments in five chains. Each chain has 2000 users and 4 nodes. We created 5000 transactions with Ethereum, Hyperledger Fabric, Fisco BCOS, and Xuperchain as the source chain and 3000 transactions with CITA as the source chain to test the impact of concurrent transactions on system stability and the impact of the number of transactions on transaction resolution time, respectively.

The transaction resolution time of ETH is shown in Figure 5. The red line indicates the average transaction resolution time, which is 6.11 s. In Figure 5, we can find that the transaction resolution time of ETH changes periodically. This is because, in the experimental environment, the average block generation time of ETH is 12 s to 14 s, which means that a transaction may be packaged at any stage of block generation, resulting in the periodic change in transaction resolution time. The right half of Figure 5 shows the dispersion of transaction resolution time.

Similarly to ETH, we mapped the transaction resolution time figures of Hyperledger Fabric, Fisco BCOS, Xuperchain, and CITA in Figure 6, Figure 7, Figure 8, and Figure 9, respectively. The red line indicates the average transaction resolution time. Based on our observation, we can learn that the cross-chain transaction resolution time of Hyperledger Fabric, Fisco BCOS, Xuperchain, and CITA is relatively stable. Except for some deviations, the transaction resolution time of the Hyperledger Fabric is between 2026 ms and 2040 ms. The transaction resolution time of Fisco BCOS is between 520 ms and 620 ms. The transaction resolution time of Xuperchain is between 1480 ms and 1620 ms. The transaction resolution time of CITA is between 19 s and 27 s. For the transaction resolution time recorded in this experiment, which includes the transaction broadcast and packaging time of the source chain, the reasons for the large difference in this part of the data are as follows: firstly, different chains have different transaction processing methods and consensus algorithms, and the transaction packaging methods of the chain itself have a direct impact on the transaction processing. Secondly, the average block time operating in different chains is slightly different.

In these figures, we can also find that the average transaction resolution time of Hyperledger Fabric is 2.032 s, the average transaction resolution time of Fisco BCOS is 1.551 s, the average transaction resolution time of Xuperchain is 0.567 s, and the average transaction resolution time of CITA is 22.729 s. Apart from the transaction processing time of each blockchain itself, this is acceptable to us. The transaction resolution time we recorded in the experiment mainly includes three parts: the transaction processing time in the source chain, transaction repackaging time, and transaction broadcast time in the Superchain. Among them, the main time consumption is the transaction processing time in the source chain. Because the transaction processing capacity of different chains is different, the data collected from different chains in the experiment are somewhat different. Therefore, apart from the necessary block-generating time of each chain, the time cost of this scheme is acceptable.

In order to verify the stability of the system, the above experiments are carried out under the condition of the dynamic joining and exiting of users, nodes, and chains. The experiment shows that the system still has stable performance under the scenario of the dynamic joining and exiting of users, nodes, and chains. At the same time, in order to further enhance the scalability of the system, we packaged the AGRobot program. If any chain belonging to the above five types wants to join the cross-chain system, it only needs to copy the AGRobot program into the server, running both the application chain node and the super chain node to join. If the new application chain does not belong to any of the five chains, it can imitate the given AGRobot program, make simple modifications, and run the program on the server running the application chain node and the super chain node at the same time to join the cross-chain system.

## 7. Conclusions

Cross-chain technology is essential for solving the problems of incompatible data formats, differences in consensus algorithms, and identity authentication and access control of heterogeneous blockchains. However, the existing cross-chain schemes have poor scalability and are without identity authentication and access control for cross-chain processes, which poses challenges to cross-chain transaction supervision and traceability. This paper proposes a cross-chain architecture with access control and identity authentication, which has high scalability and low intrusion. We theoretically proved the security of the scheme and implemented a prototype cross-chain system based on this architecture in Ethereum, Hyperledger Fabric, Fisco BCOS, Xuperchain, and CITA, and designed concurrent transaction experiments, which demonstrated the stability, scalability, and efficiency of the system under multi-users and multi-nodes.

**Future work.** We pointed out several interesting problems as our future work. As the relay chain needs to process and record the transactions from each application chain, the relay chain needs to bear a large communication and storage load. Therefore, reducing the pressure of the relay chain while ensuring the stability of the system is a problem worth studying. In addition, how to access cross-chain systems for blockchains that do not support smart contracts is also one of the future work directions. At the same time, ensuring the atomicity of cross-chain calls of smart contracts is also an important scientific issue. Therefore, our future work will focus on the three following aspects: first, reducing the communication and storage load of the relay chain while maintaining the system availability, security, and stability; second, realizing access to blockchains that does not support smart contracts; and finally, realizing the atomicity of smart contract cross-chain call. 

## Figures and Tables

**Figure 1 sensors-23-02000-f001:**
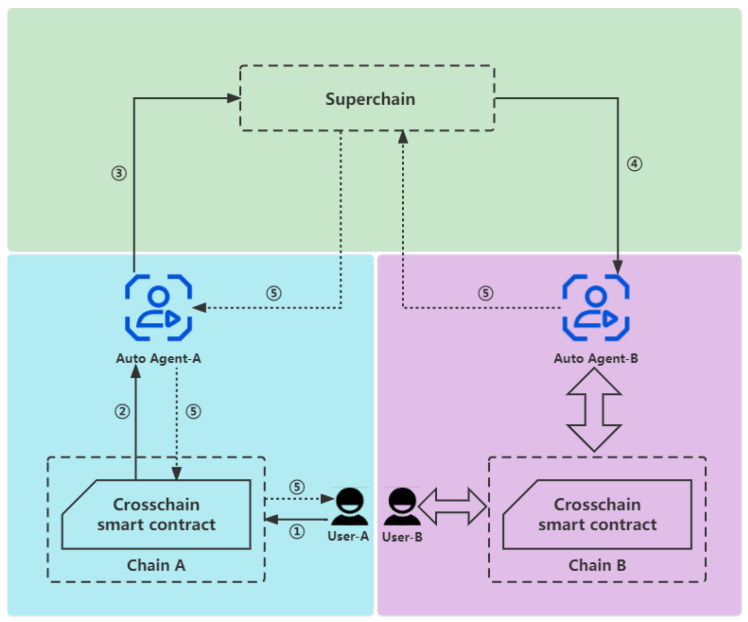
The framework of the basic cross-chain system.

**Figure 2 sensors-23-02000-f002:**
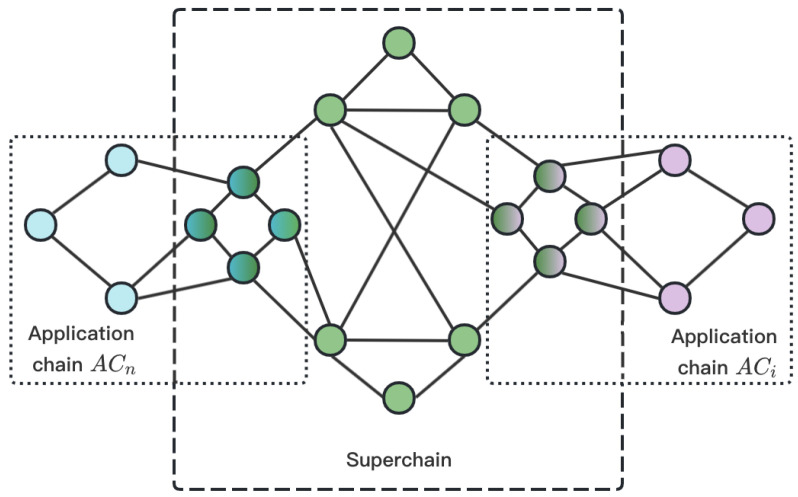
The network of the cross-chain system.

**Figure 3 sensors-23-02000-f003:**
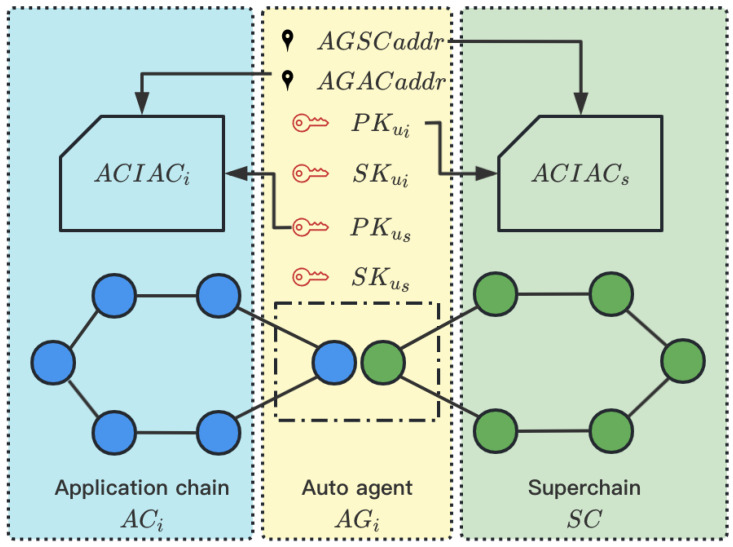
Chain Registration with Access Control and Identity Authentication.

**Figure 4 sensors-23-02000-f004:**
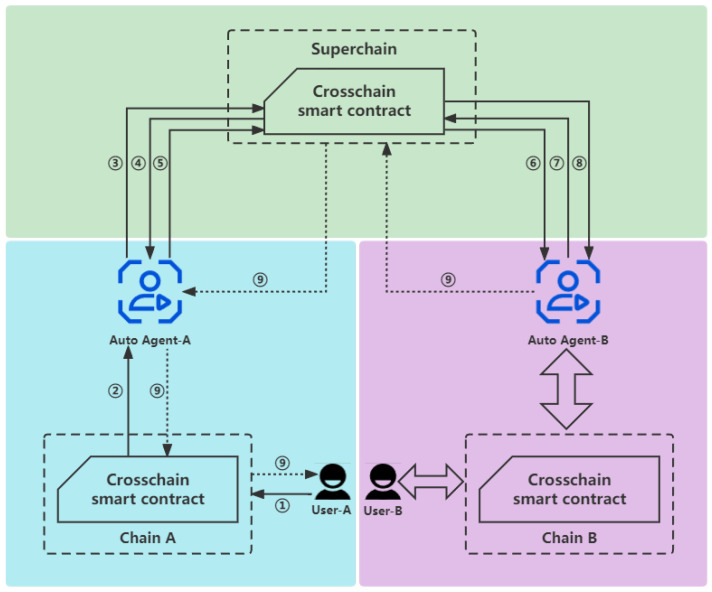
The framework of the cross-chain identity authentication and authority control scheme.

**Figure 5 sensors-23-02000-f005:**
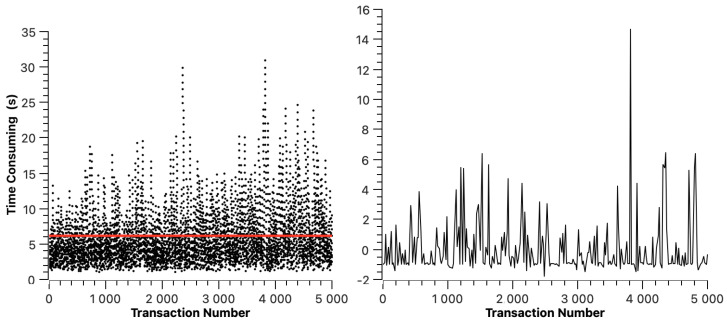
Ethereum cross-chain transaction resolution time.

**Figure 6 sensors-23-02000-f006:**
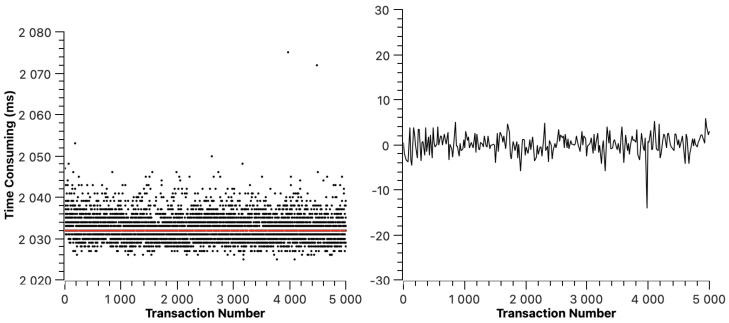
Fabric cross-chain transaction resolution time.

**Figure 7 sensors-23-02000-f007:**
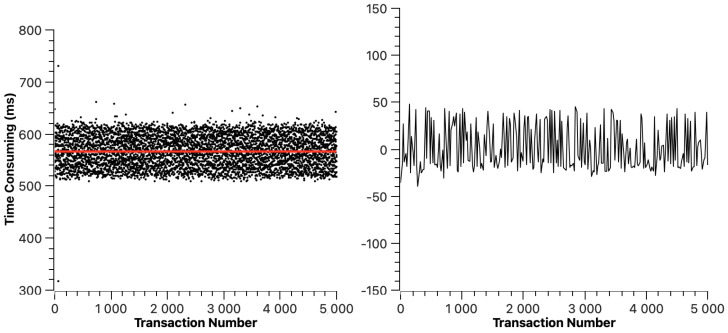
Fisco BCOS cross-chain transaction resolution time.

**Figure 8 sensors-23-02000-f008:**
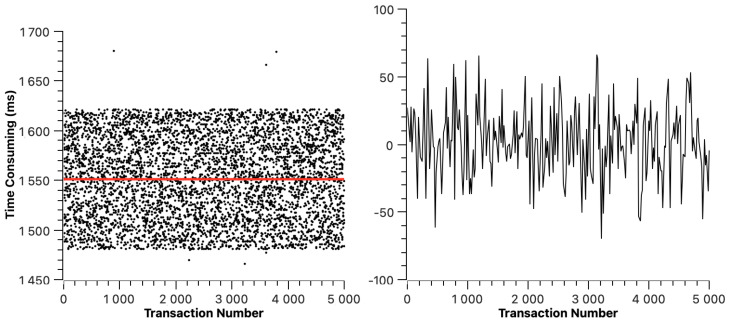
Xuperchain cross-chain transaction resolution time.

**Figure 9 sensors-23-02000-f009:**
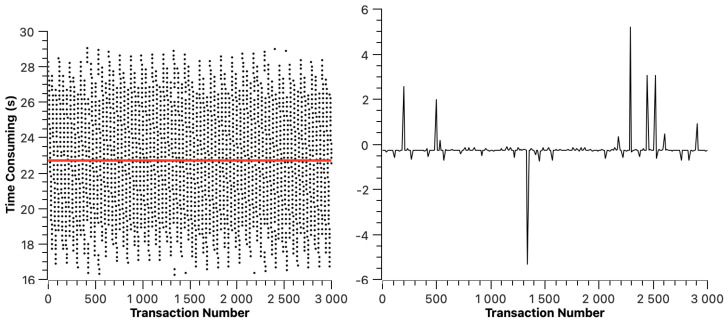
CITA cross-chain transaction resolution time.

**Table 1 sensors-23-02000-t001:** References Summary.

References	Category	Implementation	Assest Transfer	Contract Interaction	Decentralization	Scalability
[16]	notary	easy	🗸	×	×	low
[17,18]	notary	easy	🗸	×	🗸	low
[19,20]	hash-locking	medium	🗸	×	🗸	low
[12,13]	sidechain	medium	🗸	🗸	🗸	medium
[14,15,21,22,23]	relay chain	hard	🗸	🗸	🗸	high

**Table 2 sensors-23-02000-t002:** Experiment configuration.

Chain Name	IP	Users Number	Nodes Number	Go-sdk
Superchain (Fabric)	172.16.65.152	2000	4	🗸
Xuperchain	172.16.65.153	2000	4	🗸
Ethereum	172.16.65.154	2000	4	🗸
CITA	172.16.65.155	2000	4	×
Fisco BCOS	172.16.65.156	2000	4	🗸
Fabric	172.16.65.157	2000	4	🗸

## Data Availability

Not applicable.

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
