# Peer review of "A Scalable Cross-Chain Access Control and Identity Authentication Scheme"

_sensors, 2023, doi:10.3390/s23042000_

Round 1

Reviewer 1 Report

The paper focuses on the timely topic of cross-chain technology, which aims at enabling message exchange between heterogeneous blockchains (i.e., blockchains provided by different vendors).

Overall, the claims and the demonstrations of the authors are rooted on a solid experimentation testbed and are duly explained along with implementation aspects.

In the following, the major concerns regarding the manuscript are reported:

  • After the introduction, it would be useful to present a case study in detail (perhaps in a dedicated section), highlighting the research questions and the open challenges that inspired the authors.

  • A Background section with preliminary definitions, anticipating the Related Work, is advisable to let non-expert readers on the cross-chain topic, but knowledgeable of blockchain technology, understand the technological reference scenario which inspired the proposal of the paper.

  • In the Related Work section, a summary table with all the papers has to be included. In such a table, distinguishing comparison features, apt to classify the works, should be pointed out, along with the pros and cons of the surveyed approaches.

Therein, the novelties with respect to the current literature should be emphasized.

  • In the Conclusion section, future research streams must be included, that is the expected extension to the presented work.

Author Response

Dear reviewer:

Thank you for your supporting comments and constructive remarks, which have been carefully considered to further enhance the clarity and quality of the manuscript entitled with A Scalable Cross-chain Access Control and Identity Authentication Scheme for the special issue of Sensors (ISSN 1424-8220) " Cryptographic Technologies for Securing Blockchain" which belongs to the section "Communications".

We have made every effort to satisfy the recommendations of the reviewers and improved the struct of this paper. The changes in the manuscript are highlighted in red and are answered point by point in the reviewer's comments. All modifications are described in the attachment.

Once again, thank you for your time and concern to our paper processing. We look forward to hearing from you. Please do not hesitate to contact us with any other questions.

Institution and address: Renmin University of China, Beijing, 100872, P.R. China

Email: ding9949@ruc.edu.cn; bo.qin@ruc.edu.cn; 

Thanks very much for your attention to our paper.

Very sincerely yours,

Yuhang Ding, Yanran Zhang, Bo Qin, Qin Wang, Zihan Yang and Wenchang Shi

Reviewer 2 Report

1- The process explained in subsection 3.2 is not clear does it explain Figure 1 or Figure 2. Please verify.

2- Based on the paper presentation Figure 2 must be placed in page 7.

3- The novelty of the proposed model not clear can you provide a comparison with different other proposed models to show the significance of the proposed model 

Author Response

Dear reviewer:

Thank you for your supporting comments and constructive remarks, which have been carefully considered to further enhance the clarity and quality of the manuscript entitled with A Scalable Cross-chain Access Control and Identity Authentication Scheme for the special issue of Sensors (ISSN 1424-8220) " Cryptographic Technologies for Securing Blockchain" which belongs to the section "Communications".

We have made every effort to satisfy the recommendations of the reviewers and improved the struct of this paper. The changes in the manuscript are highlighted in red and blue and are answered point by point in the reviewer's comments. All modifications are described in the attachment.

Once again, thank you for your time and concern to our paper processing. We look forward to hearing from you. Please do not hesitate to contact us with any other questions.

Institution and address: Renmin University of China, Beijing, 100872, P.R. China

Email: ding9949@ruc.edu.cn; bo.qin@ruc.edu.cn; 

Thanks very much for your attention to our paper.

Very sincerely yours,

Yuhang Ding, Yanran Zhang, Bo Qin, Qin Wang, Zihan Yang and Wenchang Shi

Round 2

Reviewer 2 Report

The authors have addressed the required comments.